# Controlling Phase Purity and Texture of K_0.5_Na_0.5_NbO_3_ Thin Films by Aqueous Chemical Solution Deposition

**DOI:** 10.3390/ma12132042

**Published:** 2019-06-26

**Authors:** Nikolai Helth Gaukås, Silje Marie Dale, Trygve Magnus Ræder, Andreas Toresen, Randi Holmestad, Julia Glaum, Mari-Ann Einarsrud, Tor Grande

**Affiliations:** 1Department of Materials Science and Engineering, NTNU Norwegian University of Science and Technology, NO-7491 Trondheim, Norway; 2Department of Physics, NTNU Norwegian University of Science and Technology, NO-7491 Trondheim, Norway

**Keywords:** KNN, thin film, aqueous chemical solution deposition

## Abstract

Aqueous chemical solution deposition (CSD) of lead-free ferroelectric K_0.5_Na_0.5_NbO_3_ (KNN) thin films has a great potential for cost-effective and environmentally friendly components in microelectronics. Phase purity of KNN is, however, a persistent challenge due to the volatility of alkali metal oxides, usually countered by using excess alkali metals in the precursor solutions. Here, we report on the development of two different aqueous precursor solutions for CSD of KNN films, and we demonstrate that the decomposition process during thermal processing of the films is of detrimental importance for promoting nucleation of KNN and suppressing the formation of secondary phases. Based on thermal analysis, X-ray diffraction and IR spectroscopy of films as well as powders prepared from the solutions, it was revealed that the decomposition temperature can be controlled by chemistry resulting in phase pure KNN films. A columnar microstructure with out-of-plane texturing was observed in the phase pure KNN films, demonstrating that the microstructure is directly coupled to the thermal processing of the films.

## 1. Introduction

Piezoelectric K_0.5_Na_0.5_NbO_3_-based (KNN) ceramics have been extensively studied the last decade to replace the state-of-the-art piezoelectric materials based on PbZr_x_Ti_1−x_O_3_ (PZT) [1,2,3]. In addition to the environmental advantage of being lead-free, KNN-based ceramics benefit from having relatively high piezoelectric coefficients [4] and high Curie temperatures [1]. Moreover, KNN has shown promising results with respect to ion release and cytotoxicity tests [5,6], which make KNN a potential candidate for biomedical applications. Investigations on bulk KNN have been dominated by compositional engineering to enhance piezoelectric performance. Fabrication of KNN thin films for incorporation in electronic devices has also been investigated [7,8,9]. Preparation of thin films of KNN has successfully been performed by physical vapor deposition (PVD) [10,11] and chemical vapor deposition (CVD) [12,13]. Fabrication of KNN thin films by chemical solution deposition (CSD) has also been targeted, and 2-methoxyethanol based solutions with cation ethoxides and acetates are by far the most used, giving KNN thin films with varying degree of texture [7,14,15,16,17,18,19,20,21,22,23,24,25]. Aqueous CSD of KNN opens for more environmentally friendly precursors and chemistry [16,22,26]. A persistent challenge in CSD synthesis of KNN thin films is the formation of alkali metal deficient secondary phases, most notably K_4_Nb_6_O_17_. Measures to avoid formation of such parasitic phases are usually implemented by using an excess amount of alkali metal in the precursor solution and by optimization of the synthesis temperature [14,20,21,23].

The ferroelectric performance of thin films is strongly correlated with the microstructure. Of special interest is the ability to tailor the degree of texturing in ferroelectric thin films, since the ferroelectric performance is coupled directly to the microstructural orientation [27]. Both polycrystalline and epitaxial KNN thin films have been prepared by CSD by tailoring the thermal processing to favor either homogeneous or heterogeneous nucleation, respectively [25]. The crystallization kinetics can also be controlled chemically by altering the nucleation temperature through modification of the precursor chemistry [16,28,29].

An aqueous niobium(V) precursor solution has recently been applied in aqueous based synthesis of KNN ceramics [30,31,32]. This precursor solution has further been used to prepare KNN thin films by CSD [26]. Both polycrystalline and highly epitaxial KNN thin films were prepared on SrTiO_3_ (STO) substrates. Two distinct regions of were observed in KNN thin films on (100) oriented STO, where the majority of the films was polycrystalline with a thin epitaxial layer close to the substrate. A secondary K_2_Nb_4_O_11_ phase was observed, particularly at the surface of the films. NaCl/KCl was included in the precursor solution to form a salt flux, which suppressed the formation of K_2_Nb_4_O_11_ although it was not completely eliminated in films deposited on (100) STO. The epitaxial layer was obtained through a thermal treatment avoiding nucleation during pyrolysis (400 °C) and using a rapid heating rate up to the annealing temperature (40 K s^−1^, 800 °C).

Here, we report on developments of the aqueous CSD route to phase pure KNN thin films. A new solution based on malic acid [33] was developed, and it was demonstrated that phase purity and microstructure can be controlled by modification of the sol-gel chemistry and the thermal processing, which alter the decomposition and crystallization kinetics during pyrolysis of the films. Finally, the in-plane ferroelectric properties of the films were characterized by using interdigitated Pt electrodes.

## 2. Materials and Methods

### 2.1. Materials Synthesis

The oxalic acid-complexed solution, referred to as KNN-Ox in the following, was prepared based on the synthesis reported by Pham et al. [26]. NH_4_NbO(C_2_O_4_)_2_·xH_2_O (99.99%, Sigma-Aldrich, St. Louis, MO, USA) was dissolved in distilled water, and dried NaNO_3_ (99%, Sigma-Aldrich) and KNO_3_ (99%, Alfa Aesar, Haverhill, MA, USA) were added to the niobium solution in a molar ratio of 1:0.525:0.525 (Nb:Na:K), giving 5 mol% excess of alkali metals. The solution was stirred on a hotplate at 70 °C for 2 h. The concentration of the final solution was 0.25 M with respect to Nb.

The malic acid-complexed solution, referred to as KNN-MA in the following, was prepared based on the synthesis described by Madaro [32]. A niobic acid precipitate was dissolved in distilled water with 0.33 M DL-malic acid (99%, Sigma-Aldrich) with a molar ratio of 1:2 (Nb:malic acid) during vigorous stirring at 70 °C, and the pH of the solution was finally adjusted to 7–8 by addition of NH_3_ solution. Finally, dried NaNO_3_ (99%, Sigma-Aldrich) and KNO_3_ (99%, Alfa Aesar) were added to the malic acid-complexed niobium solution in a ratio of 1:0.525:0.525 (Nb:Na:K), giving 5 mol% excess of alkali metals. The final solution had a Nb concentration of 0.14 M.

KNN thin films were deposited on (100) oriented SrTiO_3_ single crystal (Crystal GmbH, Berlin, Germany) substrates. Prior to deposition, the substrates were rinsed with ethanol (96%), heat-treated at 550 °C for 5 min in a rapid thermal processing (RTP) oven (RTP-1, Jipelec Jetfirst 200 mm, Semco Technologies, Montpellier, France) to remove any organic residues, before the substrate surface was activated in an oxygen plasma cleaner (Femto, Diener Electronics, Ebhausen, Germany), improving the wettability. The KNN precursor solutions were then deposited on the substrates, using a spin coater (WS-400B-6NPP-LITE/AS, Laurell Technologies, Montgomery, PA, USA) at 3500 rpm for 40 s, followed by drying on a hotplate at 200 °C for 3 min. The films were then pyrolyzed at 550 °C for 5 min in either flowing O_2_ (RTP-1) or in air using a rapid heating plate (RTP-2) as described by Blichfeld et al. [34] at a heating rate of 100 K min^−1^. Deposition was repeated until a desired film thickness was obtained (15 layers deposited if not otherwise specified). Finally, the thin films were annealed at 700 °C for 10 min in O_2_ (RTP-1) or air (RTP-2), at a heating rate of 100 K min^−1^.

Powders from the two precursor solutions were prepared by drying the KNN-Ox and KNN-MA precursor solutions at 100 °C and 200 °C, respectively, for 24 h, followed by hand milling. The powders were heat-treated at various temperatures for further characterization.

### 2.2. Characterization

The thermal decomposition of the gel powders was analyzed with thermogravimetric analysis combined with mass spectroscopy (TGA-MS, TGA: STA 449C, Netzsch, MS: QMS 403C, Netzsch, Selb, Germany) up to 725 °C in flowing synthetic air and O_2_, using a heating rate of 10 K min^−1^. Fourier-transform infrared spectroscopy (FTIR, Vertex 80v, Bruker, Billerica, MA, USA), using an attenuated total reflection (ATR) cell, was performed of the gel powders heat-treated in air at various temperatures. The phase composition of the powders was determined using X-ray diffraction (XRD, D8 Advance, Bruker). The phase composition of the prepared KNN films was studied using gracing incidence X-ray diffraction (GI-XRD, D8 A25 Advance, Bruker) with an incidence angle of 2°, and the average microstructure of the films was analyzed using regular X-ray diffraction (θ–2θ XRD, D5005, Siemens, Karlsruhe, Germany) with scanning geometry parallel to the (100) direction of the substrate. The film thickness and local microstructure were studied using scanning electron microscopy (SEM, Ultra 55, Carl Zeiss AG, Oberkochen, Germany) and transmission electron microscopy (TEM, JEM-2100 and JEM-2100F, Jeol, Tokyo, Japan, both operated at 200 kV). Cross-sectional SEM samples were prepared by scribing and breaking the films using a diamond tip scriber (DX-III, Dynatex International, Santa Rosa, CA, USA). The SEM imaging was performed using an in-lens detector and an acceleration voltage of 10 kV. The TEM samples were prepared using focused ion beam (FIB, Helios Nanolab DualBeam, FEI Company, Hillsboro, OR, USA) using a standard lift-out technique. TEM imaging was performed in both bright field imaging (BFI) and high resolution (HR) mode. Fast Fourier transforms (FFTs) were applied to selected areas in the high resolution TEM (HRTEM) images to obtain information on the local crystal structure of the substrate, the thin film and the interface. Crystal orientation maps were obtained by performing scanning precession electron diffraction (SPED). The recorded data was used to calculate misorientation in the film relative to the substrate. Nanomegas ASTAR hardware and software were used for acquisition and template matching of the precession electron diffraction (PED) patterns. The template used for KNN was based on a pseudo-cubic crystal structure (data in Table A1, Appendix A). The in-plane ferroelectric response of the films was analyzed by measuring polarization-electric field (P-E) hysteresis loops, using a piezoelectric evaluation system (PES, aixPES, aixACCT, Aachen, Germany), and interdigitated Pt electrodes (electrode dimensions described in Table A2, Appendix A) deposited on top of the films. The measurements were performed at 0.1 and 0.3 Hz, using an electric field bias of 4 kV cm^−1^.

## 3. Results

### 3.1. Deposition of KNN Thin Films

KNN thin films were successfully prepared by spin coating using both precursor solutions. The solutions wetted the substrates well, resulting in dense films with uniform thickness as evident from the SEM images shown in Figure 1. The final film thickness per deposition was ~23 and ~17 nm for the KNN-Ox and KNN-MA films, respectively. The main shrinkage of the deposited films took place during pyrolysis, while the thickness remained relatively constant during the final annealing step. The grain size of the annealed films, determined from the surface images (Figure 1), varied from 30 to 170 nm, with average values of 110 and 105 nm for the KNN-Ox and KNN-MA films, respectively. Good adhesion of the films to the substrate was observed as no delamination occurred during film handling and characterization or sample preparation for TEM.

GI-XRD patterns of KNN-Ox and KNN-MA films are shown in Figure 2, demonstrating the presence of Bragg reflections of KNN for films pyrolized at 550 °C and annealed at 700 °C. The diffraction lines of the films become sharper by annealing at 700 °C. It is important to note that secondary phases (K_4_Nb_6_O_17_, K_2_Nb_4_O_11_) are present in the two KNN-Ox films and the KNN-MA film prepared in O_2_ (RTP-1), while the KNN-MA film prepared in air (RTP-2) is phase pure according to GI-XRD. The secondary phases were also present in KNN-Ox films prepared in air (RTP-2) and pyrolized at 450 °C (Figure A1, Appendix A).

θ–2θ X-ray diffraction patterns of the KNN-MA film prepared in air is presented in Figure 3, including the diffraction pattern of the STO substrate. The patterns confirm the phase purity of KNN prepared from the KNN-Ma solution and thermal treatment in air. The insets in Figure 3 highlight the (100) and (200) Bragg reflections of the KNN films. The pseudo-cubic (100), (110) and (200) reflections of KNN are clearly present for the KNN-MA film, and the relative intensity of the pseudo-cubic (100) and (110) reflections demonstrate substantial texturing. The (100):(110) intensity ratio of the KNN-MA film annealed at 700 °C is approximately 5.2, compared to the theoretical ratio of 0.67 for KNN powders.

The microstructure of the KNN-MA film annealed at 700 °C in air (RTP-2) is shown in Figure 4. The phase purity of the films was confirmed by electron microscopy and electron diffraction. The microstructure is dominated by continuous columnar KNN grains through the film. The average thickness for each deposition was 17 ± 1 nm based on electron microscopy (SEM and TEM). A thin interfacial layer of 1–3 nm was observed between the substrate and the KNN-MA film (Figure 4a). An orientation map with reconstructed grains is presented in Figure 5a. The coloring gives the misorientation relative to the substrate (blue-yellow, low-high), demonstrating a polycrystalline microstructure. This is supported by the figures presented in Figure 5b, showing no in-plane texturing in the calculated orientation distribution function (ODF) probability plot and only some clustering of crystal coordinates originating from the substrate in the pole figures.

P-E hysteresis loops of the KNN-MA film prepared in air (RTP-2) at 0.1 and 0.3 Hz are presented in Figure 6a with the current flow for these measurements in Figure 6b. Ferroelectric polarization switching is present at both frequencies, indicated by current spikes at the coercive field (~2.2–2.7 kV cm^−1^) in Figure 6b. The ferroelectric switching is obscured by leakage current. At higher frequencies (<10 HZ) ferroelectric switching was not observed, also when the applied field was increased to 120 kV cm^−1^ (Figure A2, Appendix A).

### 3.2. Preparation of KNN Powders

Powders prepared from the precursor solutions were investigated to give additional information concerning the decomposition/crystallization kinetics. XRD patterns of the two heat-treated powders from the precursor solutions (KNN-Ox and KNN-MA) are given in Figure 7. Crystallization of the KNN perovskite phase appeared at 450 °C in both powders, although the crystallinity is more prominent in the KNN-Ox powder at this temperature. Phase pure KNN powders were obtained from 500 °C and 550 °C for KNN-Ox and KNN-MA, respectively. Notably, K_4_Nb_6_O_17_ was formed at 450 °C in the KNN-Ox powder, but this phase vanished after heat treatment at 500 °C.

The thermal decomposition of the powders from the precursor solutions was further studied by TGA-MS in synthetic air and O_2_, and the data are shown in Figure 8. The KNN-Ox precursor (Figure 8a) decomposes in the temperature interval 200 to 400 °C, compared to the two-step decomposition of the KNN-MA precursor at 200 to 400 °C and 500 to 550 °C, Figure 8b. The mass loss in the KNN-Ox precursor powder before 200 °C is mainly assigned to evaporation of water, followed by an endothermic decomposition between 200 °C and 300 °C. The mass loss observed at 350–400 °C is mostly due to combustion of carbon species and decomposition of nitrates, as evident from the MS-signatures and the exothermic dip at this temperature interval. The thermogravimetric (TG) data for the KNN-Ox powders in air and O_2_ are more or less identical from 400 °C and above. The overall mass loss in the KNN-Ox precursor is ~55 wt.%. The first decomposition step (200 to 400 °C) of the KNN-MA precursor powder is analogous to the decomposition of the KNN-Ox powder between 200 and 300 °C, although the overall energy flux is exothermic in this step. The second decomposition step is an exothermic combustion of carbon and decomposition of nitrates, shifted up in temperature to 500–550 °C compared to the same decomposition in the KNN-Ox gel (350–400 °C). This decomposition step is notably steeper for the KNN-MA powder in oxygen atmosphere (500–515 °C) than in synthetic air (500–550 °C). The overall mass loss of the KNN-MA precursor is ~65 wt.%, where ~20 wt.% is lost in the second decomposition step.

Fourier-transform infrared spectra of the powders heat-treated at different temperatures, presented in Figure 9, suggest no precursor decomposition at 200 °C as the vibrational modes are mostly resembling the modes of the complexing agents, i.e., oxalic and malic acid for KNN-Ox and KNN-MA, respectively. The IR spectra show diminished and shifted vibrational modes of the initial bands for both powders after heat treatment at 300 °C. At 400 °C, only bands corresponding to Nb-O (~525 cm^−1^) are observed in the KNN-Ox powder in addition to C–O/O–C=O (~1300 cm^−1^) and C=O (~1600 cm^−1^), corresponding to carbonates. A broad band at 1100–1700 cm^−1^ in the temperature range 400–500 °C is observed in the KNN-MA powder, marked with horizontal arrows in Figure 9b. This band is confirming the presence of residuals of organics in line with the second mass loss in the TGA data (Figure 8b) and an amorphous phase at 450 °C in the powder XRD (Figure 7b). This is in coherence with the powder going from a brown to white color between 500 and 550 °C. The characteristic perovskite band at ~525 cm^−1^ show the formation of crystalline KNN after heat treatment at 500 and 550 °C for the KNN-Ox powder and KNN-MA powder, respectively.

## 4. Discussion

### 4.1. Phase Purity

Both precursor solutions yielded phase pure KNN by thermal treatment of powders from the solutions (Figure 7), which is in accordance with previous work [30,32]. However, only the KNN-MA solution heat-treated in air gave phase pure KNN thin films, as evident from the GI-XRD patterns in Figure 2. Alkali metal deficient secondary phases (K_4_Nb_6_O_17_, K_2_Nb_4_O_11_) formed in the films synthesized from the KNN-Ox and the KNN-MA solution in O_2_. Both these secondary phases have been reported previously in relation to KNN synthesis [36], however, for KNN thin film synthesis K_4_Nb_6_O_17_ is the most frequently reported due to the low nucleation temperature [20,37] and the instability of K_2_Nb_4_O_11_ at lower temperatures [35,38]. Previous work using a similar oxalic acid-complexed niobium precursor solution reported formation of the K_2_Nb_4_O_11_ secondary phase in ceramics sintered at 1100 °C [31] and in thin films heat-treated at 800 °C [26]. K_4_Nb_6_O_17_ was also observed in the 450 °C in heat-treated KNN-Ox powder (Figure 7a), but the phase was eliminated by further heat treatment at 500 °C or higher. The presence of K_4_Nb_6_O_17_ in both the KNN-Ox powder and KNN-Ox films, and the elimination of this phase in the powder and not in the films, can be explained by two phenomena: (i) Nucleation kinetics of the crystalline phases and (ii) volatilization of alkali metals. The results presented here demonstrate that both phenomena are dependent on the precursor chemistry and atmosphere during thermal treatment. More specific, the nucleation kinetics and evolution of secondary phases are dependent on the precursor decomposition temperature relative to the nucleation temperature. In general, the precursor decomposition is observed to be completed before the nucleation of KNN in the KNN-Ox system, whereas the decomposition is not yet completed when nucleation of KNN takes place in the KNN-MA system.

#### 4.1.1. Nucleation Kinetics

Nucleation of the targeted KNN phase occurred at around 450 °C in both the KNN-Ox and KNN-MA powders (Figure 7), although the crystallinity was higher in the KNN-Ox powder. Phase pure KNN was obtained at 500 °C and 550 °C in the KNN-Ox and KNN-MA powders, respectively. However, K_4_Nb_6_O_17_ was also formed at 450 °C in the KNN-Ox powder. At the temperature when KNN was first formed (450 °C), the KNN-Ox powder mostly consists of amorphous niobium oxide and alkali metal carbonates in addition to KNN, whereas the thermal decomposition of the organic residuals in the KNN-MA powder was not completed, as evident from the TGA and FTIR data (Figure 8; Figure 9, respectively). In the KNN-MA powder, the nucleation of crystalline phases was affected by the delayed decomposition process, thus shifting the nucleation temperature upwards and suppressing the formation of the K_4_Nb_6_O_17_ secondary phase. In case of the KNN-Ox powder, the nucleation of the K_4_Nb_6_O_17_ secondary phase was not suppressed in the same manner and nucleation of both KNN and K_4_Nb_6_O_17_ occurred at 450 °C. These observations from the KNN powders are in line with the GI-XRD patterns in Figure 3, as alkali metal poor secondary phases are formed in the films from the KNN-Ox precursor solution. Reduction of the pyrolysis temperature during processing of the KNN-Ox films was also attempted (Figure A1, Appendix A), but this did not affect the phase composition of the films compared to the films pyrolyzed at 550 °C. Traces of a secondary phase are also observed in the films from the KNN-MA precursor solution heat-treated in oxygen. This is expected due to a more efficient decomposition of the residual mass in the system around 500 °C, as observed in the TGA measurement (Figure 8), thus enabling nucleation of K_4_Nb_6_O_17_ during pyrolysis in oxygen. In short, elevation of the nucleation temperature, e.g., by engineering of the decomposition mechanisms of the precursor solution, is crucial to avoid nucleation of secondary phases during synthesis of KNN from solutions.

#### 4.1.2. Alkali Metal Volatility

The absence of alkali metal poor phases in the KNN-MA powder at 450 °C was rationalized by the presence of residuals of organics. The volatility of alkali metals oxides must also be taken into consideration to explain the evolution of the secondary phases in the KNN-Ox powder and thin films. As evident from the XRD in Figure 7a, the secondary phase formed at 450 °C in the KNN-Ox powder could be removed by further heat treatment at 500 °C. The disappearance of this secondary phase at higher temperatures in the powder suggests that K_4_Nb_6_O_17_ is metastable and will react with residual alkali carbonates in the powder to form KNN when the reaction kinetics (temperature) is increased. Alkali metal carbonates are present in the KNN-Ox powder at 450 °C, as seen in the KNN-Ox IR-spectrum (Figure 9a) at 400 °C where C–O/O–C=O and C=O modes are observed. The reaction involving the secondary phase can be formulated as
K_4_Nb_6_O_17_ (s) + K_2_CO_3_ (s) → 6 KNbO_3_ (s) + CO_2_ (g)(1)
where the presence of Na is neglected for simplicity.

In the case of KNN films, the surface area is significantly higher than for the powder, and diffusion of volatile species away from a flat surface with purging furnace atmosphere is higher compared to evaporation from a porous powder compact. Alkali metal carbonates are volatile and pose a constant challenge in KNN synthesis due to loss through evaporation [1,2,3]. Therefore, loss of alkali metal carbonates due to evaporation during the heat treatment of the thin films will result in irreversible changes in the cation stoichiometry, and K_4_Nb_6_O_17_ cannot react to form KNN according to Reaction (1). This demonstrates that once K_4_Nb_6_O_17_ is formed during film pyrolysis, it cannot be removed by Reaction (1) by further thermal treatment if irreversible changes in the alkali metal to niobium stoichiometry has taken place due to evaporation of alkali metal carbonates. Similar observations have been reported by Wang et al. [39], where the loss of alkali metals was suppressed by addition of organic stabilizing agents (MEA, DEA and EDTA) to the precursor solution.

### 4.2. Texture and Microstructure

The TEM characterization of the KNN-MA film synthesized in air (Figure 4 and Figure 5) revealed a 1–3 nm thick epitaxial interface layer between the substrate and the KNN film, followed by a columnar microstructure throughout the film thickness. Fast Fourier transform of the interface area (Inset II in Figure 4) give information from the combination of Areas I and III, but the TEM misorientation mapping (Figure 5a) demonstrate that this layer is epitaxial KNN (thin blue area at the bottom). The thin epitaxial layer corresponds to the inner part of the first deposited layer (17 nm). The θ–2θ diffractions of the KNN-MA films demonstrated that the films have out-of-plane texturing (Figure 3. This is due to preferential orientation of the columnar grains, despite that the PED pattern mapping indicate a local in-plane polycrystalline microstructure. No symmetry pointing towards texturing can be observed in the pole figures (Figure 5b). This is further supported by plotting misorientation of PED patterns relative to the substrate (Figure A3, Appendix A), where the misorientation follows the Mackenzie distribution of polycrystalline materials [40]. This suggests a mixed heteroepitaxial and somewhat oriented homogeneous nucleation and growth in the first deposited layer [41]. The crystallization mechanism of the subsequent layers is dominated by heterogeneous nucleation, forming a thin epitaxial interface and columnar grains with out-of-plane preferential orientation. Using a thermal treatment program with a low pyrolysis temperature (400 °C) and a high heating rate and annealing temperature (40 K s^−1^, 800 °C), Pham et al. [26] obtained highly textured KNN films using an oxalic acid-complexed niobium precursor (KNN-Ox) similar to the one presented in this work. Pyrolysis at 400 °C prevents nucleation of KNN until the final annealing, and combined with the use of a salt flux, strongly textured growth of the film was promoted. Therefore, the thermal processing during synthesis dictates the degree of texturing/epitaxy of the final films, which is in line with Yu et al. [25]. This suggests that phase pure epitaxial KNN films should be obtainable from the KNN-MA precursor solution by altering the thermal processing during synthesis

### 4.3. In-Plane Ferroelectric Strain

A ferroelectric response in the polarization-electric field measurements (Figure 6) of the KNN-MA film was only obtained at low frequencies (0.1 and 0.3 Hz) and field (4 kV cm^−1^). At higher frequencies, the film exhibited a capacitive behavior (Figure A2, Appendix A). However, at the conditions presented in Figure 6 (0.1 and 0.3 Hz, 4 kV cm^−1^), a leakage current is clearly present. The polarization observed in these measurements are expected to be due to (i) strain-induced out-of-plane preferential polarization, which is normal to the in-plane electric field used in this study and (ii) a parasitic contribution from the leakage current.

(i) Strain-induced preferential polarization: Synthesis of a thin film on a dissimilar substrate will cause misfit strain in the film due to different lattice parameters and thermal expansion coefficients in the two materials. The lattice parameter of bulk STO is smaller than the pseudo-cubic lattice parameter of KNN [42,43], and the thermal expansion coefficient of STO is higher than that of KNN [44,45]. The KNN film will thus experience compressive in-plane strain promoting out-of-plane polarization [46]. Consequently, only a minor polarization switching was observed (Figure 6). Growing the KNN films on a substrate giving tensile in-plane strain (e.g., SrRuO_3_/Pt/MgO [47], Pt/SrTiO_3_ [48]) will promote in-plane polarization and should therefore promote polarization saturation when conducting in-plane measurements.

(ii) Leakage current: Current leakage is a persistent challenge in KNN thin films due to formation of conductive electron holes (*h^•^*) in the films [49]. It is proposed that the electron holes form due to oxidation of the KNN films, enabled by the presence of oxygen vacancies in the films that originate from loss of alkali oxides (K_2_O or Na_2_O) at the film surface during synthesis. Mn-doping is usually applied in KNN to reduce the leakage current [49,50]. Dopants manipulating phase boundaries in KNN to enhance the ferroelectric properties has been thoroughly investigated by others [1] and might also be necessary to observe a stronger ferroelectric behavior in KNN thin films.

## 5. Conclusions

An environmentally friendly synthesis route to phase pure KNN films by aqueous chemical solution deposition was successfully developed. The phase purity of KNN films from two different precursor solutions, differentiated by the complexing agent for niobium, was demonstrated to be related to the decomposition temperature of the precursors during synthesis. Phase pure, dense KNN films were obtained from the solution with malic acid as complexing agent (KNN-MA) and when processed in air due to suppressed nucleation of the secondary phase due incomplete thermal decomposition of the deposited film. The decomposition of the precursors during thermal treatment of the KNN-MA films was completed at 550 °C (air) and 515 °C (O_2_), and ~20 wt.% of undecomposed mass remained at the nucleation temperature. The phase pure KNN-MA films were shown to have an out-of-plane textured columnar microstructure and to have ferroelectric properties. Finally, KNN films containing secondary phases (K_4_Nb_6_O_17_, K_2_Nb_4_O_11_) was obtained from the solution with oxalic acid as complexing agent (KNN-Ox) due to limited suppression of nucleation of the secondary phase due to a lower thermal decomposition temperature in this case. The work demonstrates the importance of chemistry for nucleation and growth and phase purity of thin films prepared by chemical solution deposition.

## Figures and Tables

**Figure 1 materials-12-02042-f001:**
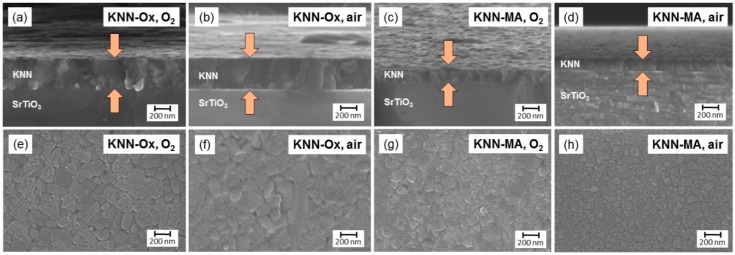
SEM micrographs of KNN films prepared by spin coating of the two precursor solutions (**a**,**b**,**e**,**f**) KNN-Ox (15 depositions) and (**c**,**d**,**g**,**h**) KNN-MA (10 depositions). (**a**–**d**) are cross-sectional images of the thin films. (**e**–**h**) are top-view images of the surface of the films.

**Figure 2 materials-12-02042-f002:**
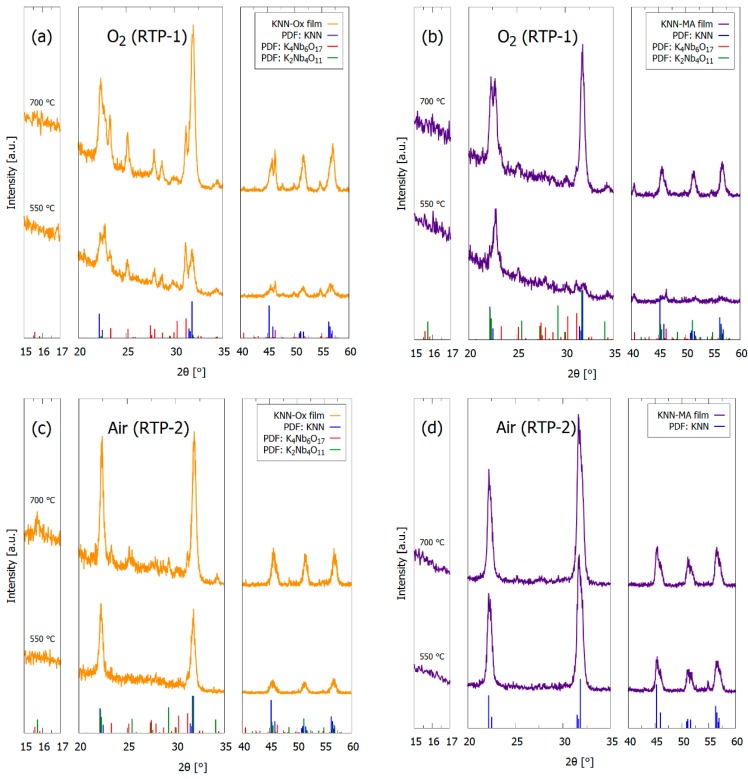
GI-XRD patterns of pyrolyzed (550 °C) and annealed (700 °C) 15-layer thick films from the precursor solutions (**a**), (**c**) KNN-Ox and (**b**), (**d**) KNN-MA. (**a**,**b**) are films thermally processed in O_2_ (RTP-1) and (**c**,**d**) are films prepared in air (RTP-2). An incident angle of 2° was used. Reference patterns for K_0.5_Na_0.5_NbO_3_ (blue, PDF card 00-061-0315), K_4_Nb_6_O_17_ (red, PDF card 04-009-6408) and K_2_Nb_4_O_11_ (green, from Madaro et al. [35]) are included in the figure.

**Figure 3 materials-12-02042-f003:**
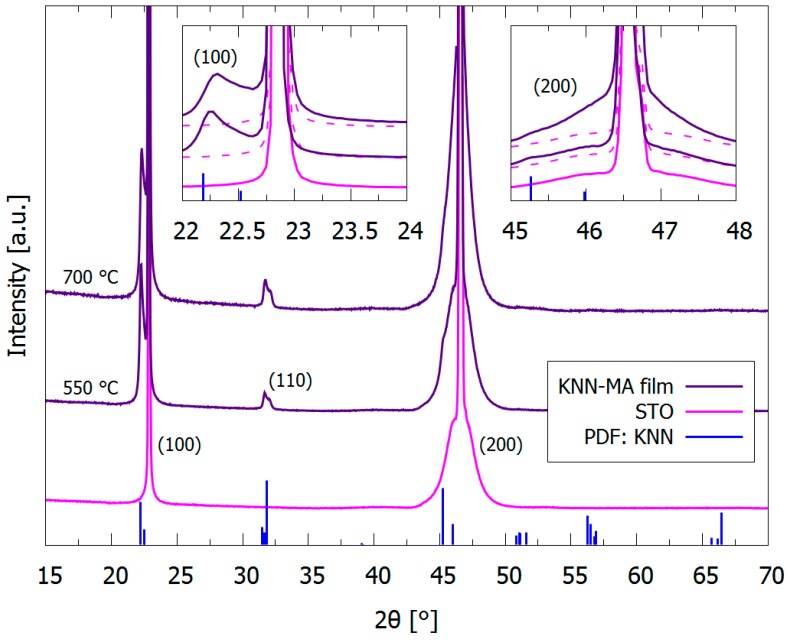
θ–2θ XRD patterns of pyrolyzed (550 °C) and annealed (700 °C) 15 layer thick KNN films prepared from the precursor solution KNN-MA and thermally processed in air (RTP-2). The insets highlight the pseudo-cubic (100) and (200) KNN reflections close to the corresponding STO reflections. The diffraction pattern of the (100) STO substrate is included. Reference pattern for K_0.5_Na_0.5_NbO_3_ (blue, PDF card 00-061-0315) is included.

**Figure 4 materials-12-02042-f004:**
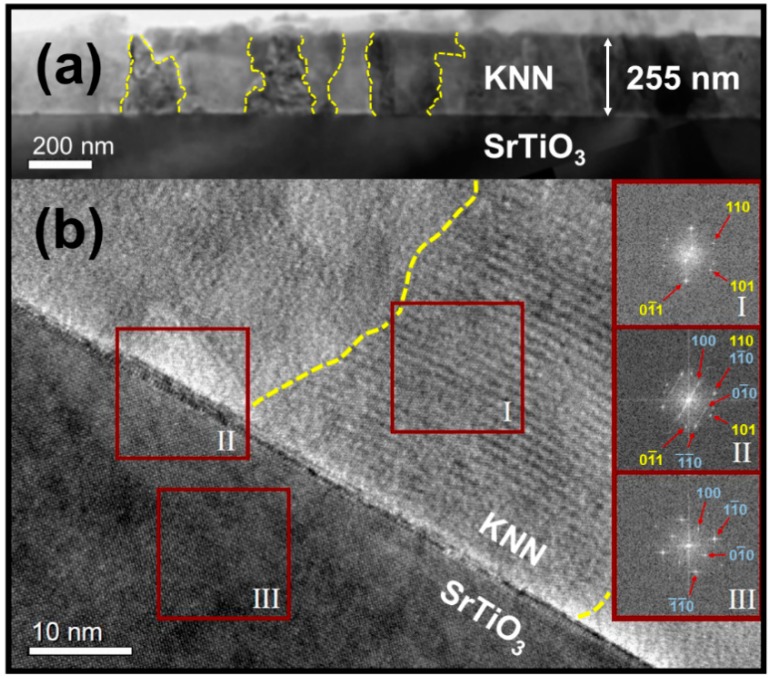
TEM images of the KNN film from the KNN-MA precursor solution prepared in air (RTP-2). (**a**) Bright field image (BFI) of the thin film cross section. Grain boundaries are highlighted with yellow lines. (**b**) High resolution TEM image (HRTEM) from the cross section with electron diffraction patterns from three selected areas (I–III). The electron diffraction pattern from Area II also contained the diffraction spots from Areas I and III.

**Figure 5 materials-12-02042-f005:**
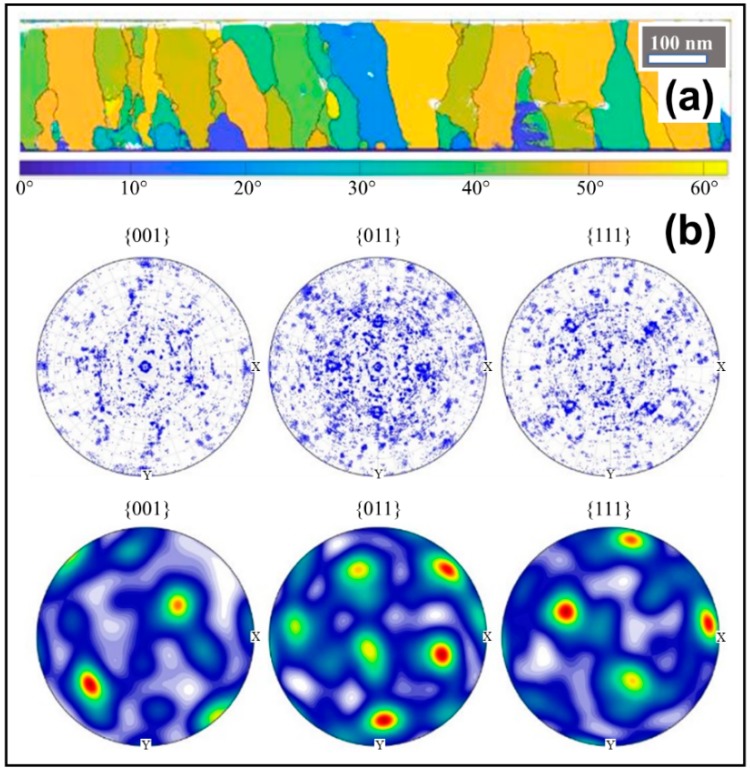
Misorientation mapping of 15-layer thick KNN film pyrolyzed at 550 °C and annealed at 700 °C in air (RTP-2), from the KNN-MA precursor solution; (**a**) Map of misorientation relative to the mean orientation of the substrate, with reconstructed grain boundaries. The substrate is located at the bottom; (**b**) Oriented pole figures of crystal coordinates (**top**), and calculated ODFs with colors indicating the probability density (**bottom**).

**Figure 6 materials-12-02042-f006:**
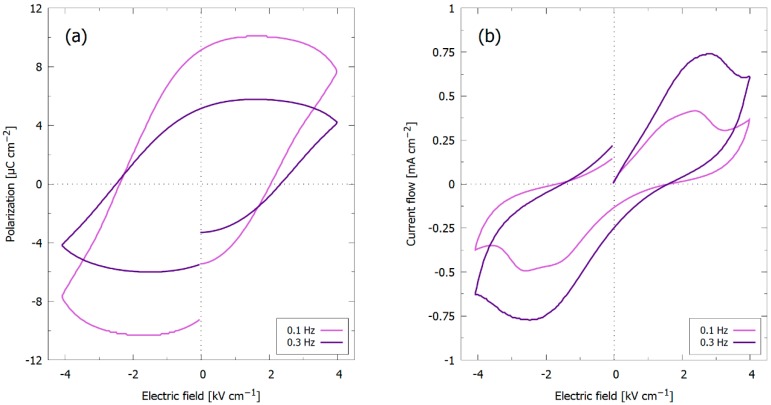
(**a**) P-E hysteresis loops of KNN-MA films prepared in air (RTP-2) at 0.1 and 0.3 Hz, using an electric field bias of 4 kV cm^−1^; (**b**) The correspond current flows during the measurements.

**Figure 7 materials-12-02042-f007:**
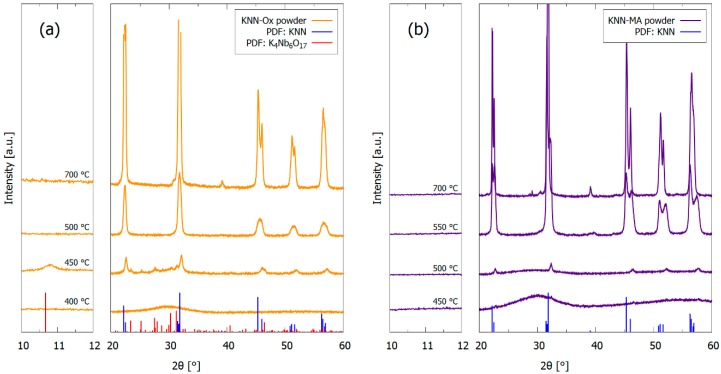
Powder X-ray diffraction patterns of heat-treated powder from the precursor solutions (**a**) KNN-Ox and (**b**) KNN-MA. Reference patterns for K_0.5_Na_0.5_NbO_3_ (blue, PDF card 00-061-0315) and K_4_Nb_6_O_17_ (red, PDF card 04-009-6408) are included.

**Figure 8 materials-12-02042-f008:**
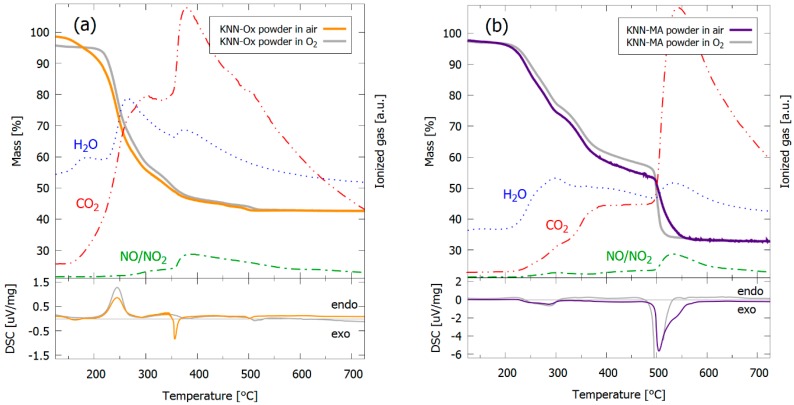
Thermogravimetric analyses of powders from the precursor solutions (**a**) KNN-Ox and (**b**) KNN-MA. The top panels contain mass loss curves (solid) from the TG and gas signatures (dotted) from the MS. The gas signatures are from the measurements in synthetic air. The lower panels contain the differential scanning calorimetry (DSC) signals from the measurements.

**Figure 9 materials-12-02042-f009:**
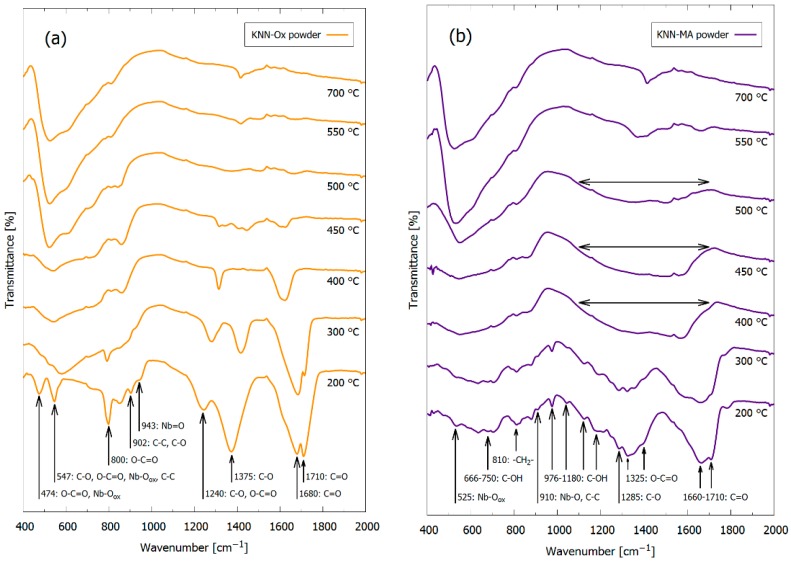
FTIR spectra of heat-treated powders from the precursor solutions (**a**) KNN-Ox and (**b**) KNN-MA. Characteristic vibrational modes are assigned. The horizontal arrows in (**b**) highlight the broad band at 1100–1700 cm^−1^, corresponding to residuals of organic species.

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
