# Peer review of "Controlling Phase Purity and Texture of K0.5Na0.5NbO3 Thin Films by Aqueous Chemical Solution Deposition"

_materials, 2019, doi:10.3390/ma12132042_

Round 1
Reviewer 1 Report
This manuscript presents the thin film growth of phase pure KNN by CSD. The authors report that phase pure and dense KNN films were obtained from the solution with malic acid as complexing agent while the solution with oxalic acid as a complexing agent (KNN-Ox) resulted in KNN films with secondary phases. The results were explained by the suppression of nucleation of the secondary phase driven by higher decomposition temperature. In comparison with common approaches of using an excess amount of alkali metal in the precursor solution, it is claimed that the presented method will provide an environmentally friendly route with desired thin film structures.
1) The authors need to comment about the use of different concentration of KNN-Ox (0.25 M with respect to Nb) and KNN-MA (0.14 M) since solution concentration is known to influence decomposition behavior as well as nucleation event. The influence of solution concentration should be explained with supporting data. If the used concentration is to obtain the same thickness per deposition, the concentration should be adjusted to obtain the same thickness for comparison. Section 3.1 describes ~ 23 nm for the KNN-Ox and ~17 nm for the KNN-MA per deposition. Thicker films may cause incomplete decomposition compared to thinner films. The authors need to clearly describe film thickness after each coating and pyrolysis, and final film thickness after annealing.
2) Page 2, Line 92: “Deposition was repeated until a desired film thickness was obtained.” What is the film thickness after annealing? Fig 1 shows about ~200 nm. Does it mean about 10 times of coating and pyrolysis?
3) It is likely that higher amounts of organic residue from KNN-MA route after pyrolysis restrict the formation of the secondary phase. If the pyrolysis temperature is lowered in KNN-Ox to contain organic residue in the pyrolyzed film, does the film show the restricted formation of the secondary phase?
Reviewer 2 Report
The work contains a lot of interesting information about the features of the technology of obtaining KNN from aqueous solutions. The results of structural and morphological studies are quite complete. However, the results of electrophysical studies are very poor, which does not allow to fully appreciate the significance of the work.
Only one of the two electrical characteristics graphs (hysteresis loop) is an independent result. Information about leakage currents and the frequency dependence of the ferroelectric properties is limited to unjustified statements. It is necessary to expand this section of work.
Round 2
Reviewer 1 Report
The authors reasonably addressed the comments raised by this reviewer. Acceptance of the manuscript for the publication is recommended.
Author Response
No further revisoon required.
Reviewer 2 Report
To illustrate the effect of leakage current, you provide dP/dE versus E graphs (Line 422. "The insets show the current flows during the measurements, illustrated as the derivative of the polarization with respect to field."). Current flows measured in nF/cm units are looking strange.
Well, let's define what the leakage current is. The leakage current is a through-current between two capacitor plates. Whatever the nature of this conductivity, its magnitude can only decrease with increasing voltage switching frequency.
On the contrary, the effect of the series resistance of the capacitor switching circuit increases with increasing frequency and, if the value of resistance is high, leads to the drop of the capacitor switching.
Let's not confuse the concepts of leakage current and switching current.
And in line 363. I do not understand completely, what does it mean:
"It is proposed that the electron holes form due to oxidation enabled by oxygen vacancies formed due to loss of alkali cations."
Round 3
Reviewer 2 Report
Thank you. Now the rising of the current at high voltages is clearly visible.
line 346. "This proposed be related to the increase in sample conductivity with increasing frequency."
I believe that "the increase in conductivity" is not wrong definition, but it is not a very good definition in the case. Conductivity is usually directly associated with conductance that is a real part of the admittance of the capacitor. But the rise of the conductance you observe at high voltages only. The rise of the current with the frequency observed in fig.A2(a) is mainly due to the susceptance B=j*ω*C (the imaginary part of admittance). You can see that the rise with the frequency is close to linear. If you would remove this sentence, for example, the discussion should become more clear.
Author Response
The sentence is removed in accordance with the comment from the reviewer